# Opalescence and Fluorescence of 46 Resin-Based Composites Exposed to Ultraviolet Light

**DOI:** 10.3390/ma17194843

**Published:** 2024-09-30

**Authors:** Soheil Ghaffari, Anubhav Gulati, Richard Bengt Price

**Affiliations:** Department of Dental Clinical Sciences, Dalhousie University, Halifax, NS B3H 4R2, Canada; s.ghaffari@dal.ca (S.G.); an623775@dal.ca (A.G.)

**Keywords:** spectrometer, light emittance, brightness, resin monomer, opalescence, fluorescence

## Abstract

Identifying the boundary between the tooth and the resin-based composite (RBC) is difficult when replacing restorations. Ultraviolet (UV) light has been reported to assist the viewer by causing the RBC to fluoresce. Using a laboratory-grade fiberoptic spectrometer, 46 RBCs were exposed to UV light from the Woodpecker O-Star curing light. The opalescence and fluorescence were measured relative to a human tooth that contained just dentin and a tooth that contained both enamel and dentin. After these quantitative measurements, 10 RBCs with large differences in light emittance were compared qualitatively to assess their brightness when exposed to UV light compared to the dentin specimen and the specimen containing both enamel and dentin. It was found that, when exposed to UV light, some of the RBCs were less bright compared to the two samples of teeth used for comparison, but most were brighter; some were up to six times brighter. The filler appears to affect the opalescence peaks, while the resin appears to affect the fluorescence peaks. It was concluded that because RBCs emit very different levels of opalescence and fluorescence, UV light from the Woodpecker O-Star cannot be relied upon to detect all brands of RBC on the tooth. The opalescence and fluorescence can also be used to detect changes in the formulation of the RBC.

## 1. Introduction

The World Health Organization has reported that untreated dental caries affects at least 2.3 billion people annually, making dental caries the most common disease globally [1]. In 2014 alone, more than 800 million direct resin-based composite (RBC) restorations were placed [2]. This number will have increased since then because there is a global agreement to phase down the use of dental amalgam, and currently RBCs are the most commonly used alternative material [3]. While filling dental cavities is vital for maintaining oral health, these restorations are not permanent [4,5,6]. It has been estimated that replacing failed restorations accounts for more than half (57%) of the restorations placed by dentists [7], and every time a restoration is replaced, more tooth structure is removed [8,9], thus accelerating the re-restoration cycle [10].

The shortened longevity of RBC restorations due to secondary caries, fracture, sensitivity, or wear means that they will have to be replaced more often than amalgam restorations [4,5,6]. One of the challenges when replacing these tooth-coloured RBC restorations is identifying the RBC on the tooth because RBCs are designed to match the colour of the original tooth [8,9]. When only visual inspection is used by dentists to identify teeth with resin restoration, the accuracy and specificity are only 31.6% and 85%, respectively [11]. To help the dentist identify the RBC on the tooth, some curing light manufacturers have included a composite detection mode in their curing light. This mode emits ultraviolet (UV) light (395 nm to 405 nm) to induce the fluorescence of the RBC [12]. This has been reported to help the dentist detect the RBC on the tooth [12,13,14] and improves the detection accuracy to 92% and the specificity to 99% [11]. But can the UV light from a dental LCU be relied upon to detect all RBCs?

The fluorescence in teeth and RBCs is attributed to the presence of compounds with highly conjugated systems that contain delocalized electrons [15,16]. The delocalized electrons lower the energy gap between molecular orbitals, making it easier for them to absorb light in the visible or ultraviolet range. These electrons are excited by incoming electromagnetic radiation (light), and when they return to their ground state, they re-emit this energy as fluorescence [16,17]. In natural teeth, dentin fluoresces up to three times more than enamel due to the presence of tryptophan in the collagen fibres. Dental manufacturers design their RBCs to closely mimic the fluorescence of natural teeth by incorporating rare earth luminescent materials (fluorophores), such as oxides of europium, cerium, and ytterbium. Even uranium oxide was used, but it was discontinued due to concerns about radioactivity [15]. However, the level of fluorescence can vary significantly depending on the type and amount of these elements incorporated into the RBC.

A separate effect is the opalescence of the RBC. Opalescence describes the extent of refraction and scattering of light within the RBC. This is impacted by the various resin matrices and the inorganic filler (microfill, nanofill, hybrid, etc.) content [16]. While fluorescence results in the emission of wavelengths longer than those from the curing light, the scattered light emitted in opalescence is at the same frequency and wavelength as the light received. Companies that make RBCs add inorganic fillers such as ZrO_2_, Al_2_O_3_, and TiO_2_, hybridized with various organic monomers to match the opalescence qualities of natural teeth.

Since no study has yet looked at a wide range of old and contemporary RBCs under controlled conditions, this study examined both quantitatively and qualitatively the effect of using the RBC detection setting on the Woodpecker O Star curing light (Guilin Medical Instrument Co., Guangxi, China) on the opalescence and fluorescence of a wide range of RBCs. The hypothesis is that, when exposed to UV light, all the RBCs will emit a similar level of opalescence and fluorescence.

## 2. Materials and Methods

### 2.1. Sample Preparation

The 46 RBCs evaluated are listed in Table 1. Unfilled resins of PowerFill (Ivoclar, Schaan, Liechtenstein), EsthetX (Dentsply Sirona, Charlotte, NC, USA), and Vit-l-escence (Ultradent, South Jordan, UT, USA) were also provided by their respective manufacturers to determine the effect of the fillers in these RBCs. Four samples of each RBC were prepared in moulds that were 5 mm in diameter and 2 mm deep (Figure 1A). A clear mylar strip (Patterson, Saint Paul, MN, USA) was placed on the bottom, allowing one side of the RBC to have a smooth, reflective surface. The top surface was flattened but remained exposed to air upon light curing. Each side wasphotocured for 20 s at a 0 mm distance using a Valo cordless light (Ultradent) emitting 1000 mW/cm^2^ on its regular power setting. This irradiance was measured using a calibrated laboratory-grade spectrometer (Ocean Optics, Orlando, FL, USA) attached to an integrating sphere.

### 2.2. Light Measurements

The opalescence and fluorescence were measured using a laboratory-grade STS-VIS Miniature Spectrometer attached to a 3.9 mm diameter cosine-corrected CC3 detector (Ocean Optics, Orlando, FL, USA). This CC3 detector was positioned no more than 0.5 mm away from the edge of the RBC (Figure 1A). UV light from the Woodpecker O-Star (Woodpecker, Guilin, Guangxi, China) on its composite detection UV setting was directed into the sample from 20 mm above the RBC, as shown in Figure 1B. This setting produced violet light with a high spectral radiant power peaking at 402 nm (Figure 1C), which is meant to detect the RBC.

### 2.3. Absolute Fluorescence and Opalescence Calculations

The four spectra for each trial were averaged to determine the opalescence and fluorescence of the RBCs. The area of the curve was then obtained from 350 to 435 nm (Equation (1): opalescence) and 435 to 700 nm (Equation (2): fluorescence) by multiplying the differences between the wavelengths ($\lambda_{n+1}-\lambda_{n}$) by the spectral radiant power (SRP) and adding up the values. No light was emitted from the LCU above 435 nm (Figure 1C). Thus, the 435 nm point was chosen as the transition point between the opalescence, which peaked at the same wavelength that was added to the system (@~402 nm), and the fluorescence, which had multiple emission peaks beyond 435 nm (Figure 2). This produced both the opalescence and fluorescence values for each RBC.
(1)$$\begin{array}{cc} Opalescence\; & \sum_{n=1}^{435}\left((\lambda_{n+1}-\lambda_{n}\right)\times {SRP}_{n}) \end{array}$$
(2)$$\begin{array}{cc} Fluorescence\; & \sum_{n=435}^{700}\left((\lambda_{n+1}-\lambda_{n}\right)\times {SRP}_{n}) \end{array}$$

### 2.4. Relative Tooth Calculations and Ranking

The opalescence and fluorescence of human teeth (REB #: 2021-5703) that matched the dimensions of the RBCs was measured using the same 435 nm cut-off wavelength between opalescence and fluorescence (Figure 2). Two tooth samples were chosen to have the most consistent colour and light emittance from these four teeth. One of the tooth samples was mostly enamel, and the other was mostly dentin. This was determined by viewing the tooth samples under regular and UV light (the dentin sample being more yellow and green under the respective lights). This tooth was used as the baseline [1] when comparing all the RBCs (Figure 3). The opalescence and fluorescence values from these teeth (Figure 3) were used to obtain the relative opalescence (Equation (3)) and relative fluorescence (Equation (4)) of each RBC. This value was used to obtain the relative light emittance (Equation (5)), which was then used to rank the RBCs.
(3)$$\frac{Absolute\;Opalescence}{Tooth\;Opalescence}\times 100\%=Relative\;Opalescence$$
(4)$$\frac{Absolute\;Fluorescence}{Tooth\;Fluorescence}\times 100\%=Relative\;Fluorescence$$
(5)$$\frac{\left[Opalescence\;\left(RBC\right)+FLuorescence\;\left(RBC\right)\right]}{\left[Opalescence\;\left(tooth\right)+Fluorescence\;\left(tooth\right)\right]}=Relative\;Light\;Emittance$$

### 2.5. Double-Blind Visual Ranking of the Brightness

After the quantitative measurements had been made using the spectrometer, 10 RBCs that had large differences in their absolute light emittance were compared qualitatively by three human evaluators. The RBCs were ranked twice, once against the relative ranking of the dentin sample, and then again against the relative ranking of the enamel-plus-dentin sample. The trained evaluators subjectively ranked the brightness of the RBCs compared to the dentin or to the enamel-plus-dentin samples when exposed to UV light. Each RBC was compared with spectrometer results to determine which RBC was brighter and which had been objectively measured to be duller than the tooth. Both the administrator of the comparisons and the three evaluators were blinded regarding what was being viewed, and the samples were randomly assigned before being placed next to one another under the UV light. The order in which the pairs were compared was also randomized.

## 3. Results

The human tooth displayed very little fluorescence compared to the RBCs (Figure 3B).

The relative light emittances of the RBCs were compared to the two specimens of human teeth [1] and ranked from least bright to most bright under the UV light from the Woodpecker O-Star (Figure 4). This calculation included the results of the specimens’ opalescence and fluorescence. RBCs that appeared closer to the tooth on the list appeared to be most like the tooth specimens when examined under UV light, while those that were furthest away appeared either darker or lighter. When exposed to UV light, five brands of RBC were more than 6× brighter than the human tooth, and five brands were less bright than the human tooth.

The three rankers unanimously ranked the samples in exactly the same order. In the blind ranking for the enamel tooth comparison, seven of the nine trials matched the light emittance ranking obtained using the laboratory-grade spectrometer (Ocean Optics). Furthermore, six of the nine trials matched the relative light emittance rankings for the tooth sample that was mostly dentin.

The RBCs behaved very differently and produced very different spectra when they were exposed to the UV light from the Woodpecker O-Star curing light that delivered a peak emission at ~402 nm. Figure 5A shows one of the brightest samples, Admira Fusion X-tra (AFX 2019, Voco, Cuxhaven, Germany, expiry date 2019), and one of the dullest samples, Activa Bioactive Restorative (ABR), in relation to the tooth specimens. The emission spectrum from AFX 2019 had multiple fluorescence peaks, while ABR had less fluorescence than the tooth sample (Figure 5B) when exposed to the violet light from the Woodpecker O-Star. Figure 5 also shows how the PowerFill monomer and the PowerFill monomer with its filler behaved. Using the monomer alone produced virtually no opalescence but a high fluorescence. The fluorescent properties decreased in the filled RBC, and the opalescence increased (Figure 5B).

There was a similar trend with the other products from Ivoclar (Figure 6A,B) and the plain resin from Dentsply Sirona (Figure 6C). For all three materials, the opalescence peak was observed only when the UV light was shone on the sample containing the monomer with filler and not the plain resin. However, this was not the case for the Vit-l-escence RBC (Figure 6D). This product had an opalescence peak even when only the plain resin received UV light.

The samples were made and tested randomly. By chance, two different lot numbers with different expiry dates of Admira Fusion X-tra were tested. When comparing the old (≤2019) expiry date and the new (≥2023) expiry date of Admira Fusion X-tra, there was a noticeable difference in the brightness and spectra. Under UV light, the newer version of Admira Fusion X-tra fluoresced less and appeared closer to a natural tooth (Figure 7B,C). In comparison, the older version of Admira Fusion X-tra was the second brightest RBC after GUI, approximately six times brighter than the natural tooth. When the sample tooth was embedded and surrounded by the two versions of the same brand of RBC (Figure 7A,B), the difference between the fluorescence under UV light was diminished, but still evident compared to individual samples (Figure 5A).

## 4. Discussion

The hypothesis for this study was rejected because the opalescence and fluorescence levels were markedly different between the RBCs when they were exposed to light from the Woodpecker O-Star curing light. Some RBCs could not be detected using this UV light.

The relative light emittance was lower for 13 RBCs compared to the dentin sample and 10 RBCs for the enamel-plus-dentin sample. However, a few RBCs were up to six times brighter than the samples of human teeth. The highest relative light emittance was observed for the GUI RBC, whereas the lowest relative light was observed from CC (Figure 4A,B).

Since every time a restoration is replaced, more tooth structure is removed [8,9], the dentist must have tools to help them distinguish between the resin and the tooth. The ranking of the 46 RBCs reported in Figure 4 can be used to establish which RBC could be most readily differentiated from the natural tooth (Figure 4A,B). By using RBCs that are brighter than or duller than teeth, a dentist could remove old restorations with less removal of natural tooth structure. At the same time, if the goal of a dentist is to match the restoration colour to the natural tooth, they would pick an RBC closer to the tooth enamel or dentin on the ranking (Figure 4A,B). A dentist may decide that posterior RBC restorations should appear more different from teeth under UV light since they are not in the esthetic zone and are replaced more frequently. In contrast, anterior restorations should be more like the tooth so that restorations are invisible if the patient is in a setting with more UV light, e.g., a nightclub. It is important to consider that if the surrounding teeth are brighter than the sample (Figure 7B), the scattering of light can blur the boundaries between tooth and composite. This is less of an issue when a bright composite is surrounded by a tooth (in a patient’s mouth); however, darker composites can receive light scattering from the tooth, which makes them less visible under UV light.

Some variables important to control for to avoid affecting the results include the thickness of the samples as well as the wavelength of the light. Previous research has reported that the thickness affects the results and that thin samples of RBC have more fluorescence and opalescence; therefore, to evaluate different RBCs, they were made in identically sized moulds. Knowing what wavelengths of light are best for eliciting a different fluorescence between natural tooth and composite resins is also important. The study conducted by Jeong et al. showed that the optimal wavelength of light for showing the apparent difference between the resin and the tooth was 405 nm, the shortest of the wavelengths measured [17]. A different study reported that UV light at 395 nm is better and more efficient at detecting compounds than white light, leaving fewer resin residues when debonding orthodontic brackets [12]. Consequently, UV light around 395–405 nm should induce fluorescence in the RBC. The Woodpecker O-Star has a pronounced peak wavelength emission at ~402 nm and that should make it ideal for detecting RBCs. However, only one LCU was used in this study and further studies are required to determine if a different peak wavelength, e.g., 395 or 410 nm would be better at detecting the RBC on enamel and dentine.

When isolating the contribution of various materials to fluorescence and opalescence, Figure 4 shows that it is impossible to make an overarching statement that UV light can detect all the RBCs on the market. Manufacturers use different resins and fillers in their RBCs, and they do not fully disclose the proprietary compositions of the base resins or fillers they use in their RBCs. Thus, the use of the RBC detection mode from the Woodpecker O-Star light cannot be relied upon to detect all RBCs. For example, based on the evaluations of the three unfilled resins provided by Ivoclar, Dentsply, and Ultradent, it appears that their proprietary resin monomers produce the fluorescence peaks under UV light in both the Ivoclar and Dentsply Sirona RBCs (Figure 6A–C), but not for the Vit-l-escence resin from Ultradent, which produced less fluorescence than its filled version (Figure 6D). Adding inorganic filler in all the RBCs tested produced opalescence peaks under UV light. However, the fluorescence cannot be attributed to any one monomer (Bis-GMA, TEGDMA, etc.) or camphorquinone since the percentages of these compounds in the dental RBCs are unknown and are proprietary trade secrets. Future studies could use controlled amounts of various known monomers and fillers to identify the cause of fluorescence and opalescence in these experimental RBCs.

It is noteworthy that changes in fluorescence could distinguish the differences between RBCs of the same brand that had been manufactured at separate times. There was a significant difference between the emission spectra of Admira Fusion X-tra that expired before 2019 and material that had yet to expire (Figure 7B,C). Thus, when exposed to the violet light from the Woodpecker O-Star, the emission spectra from this RBC showed that there had been a change in its composition that was not public knowledge.

## 5. Conclusions

When exposed to UV light from the Woodpecker O-Star curing light, the RBCs emitted very different levels of opalescence and fluorescence. Some RBCs could not be detected using this UV light.The filler used in the Ivoclar and Dentsply Sirona RBCs affects the opalescence peaks, while the resin monomer influences the fluorescence peaks.Emission spectra from RBCs exposed to violet light can differ depending on when the RBC was manufactured. This can disclose changes that the manufacturer made to the composition of the RBC.

## Figures and Tables

**Figure 1 materials-17-04843-f001:**
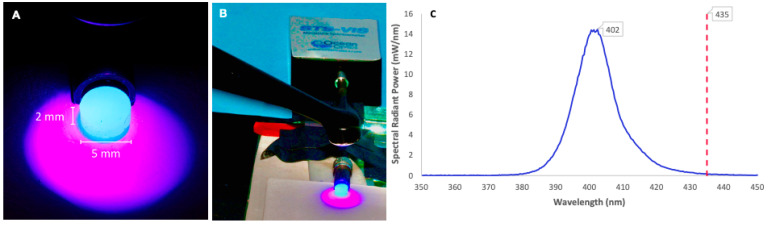
(**A**) The distance of the resin-based composite sample was between 0 and 0.5 mm from the CC3 detector (behind). (**B**) The Woodpecker O-Star curing light was fixed 20 mm above the RBC. The sample was centred under the UV spot. (**C**) The spectral radiant power (mW/nm) of the light from the Woodpecker O-Star. Emission peak at 402 nm. Beyond 435 nm, no light was detected.

**Figure 2 materials-17-04843-f002:**
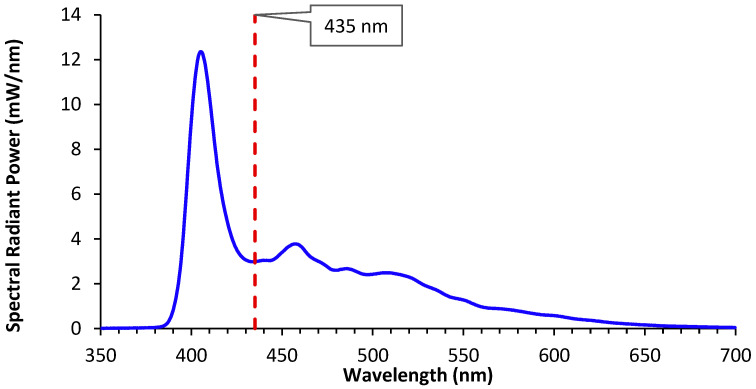
Average spectra from 4 samples of Admira Fusion X-tra with a 2023 expiry date. The dotted line indicates the point at which the spectrum transitioned from opalescence, scattering the same wavelength delivered, to fluorescence, which produced wavelengths of light not coming from the curing light.

**Figure 3 materials-17-04843-f003:**
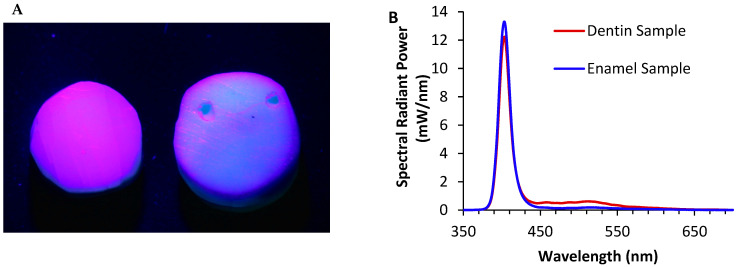
(**A**) Samples of the two human teeth that exhibited the most consistent colour and light emittance. The sample on the left was mostly enamel, and the sample on the right was mostly dentin. (**B**) Reflected spectra from the same two tooth samples showing relatively similar wavelength peaks.

**Figure 4 materials-17-04843-f004:**
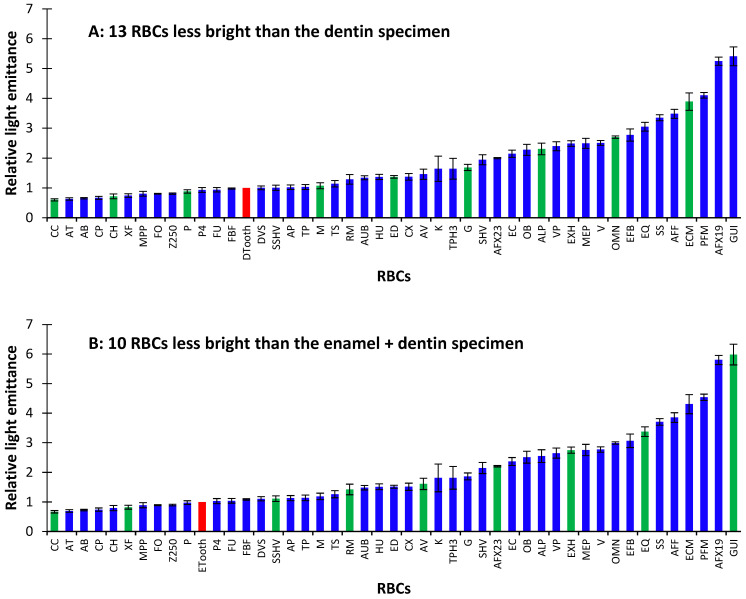
Ranking of the light emittance from the resin-based composites (RBCs) compared to the human tooth. (**A**) Relative light emittance compared to a tooth sample composed mainly of dentin. (**B**) Relative light emittance to a tooth sample composed mainly of enamel. The tooth sample is highlighted in red in each figure. The samples of the RBCs that were used in the blind ranking are highlighted in green.

**Figure 5 materials-17-04843-f005:**
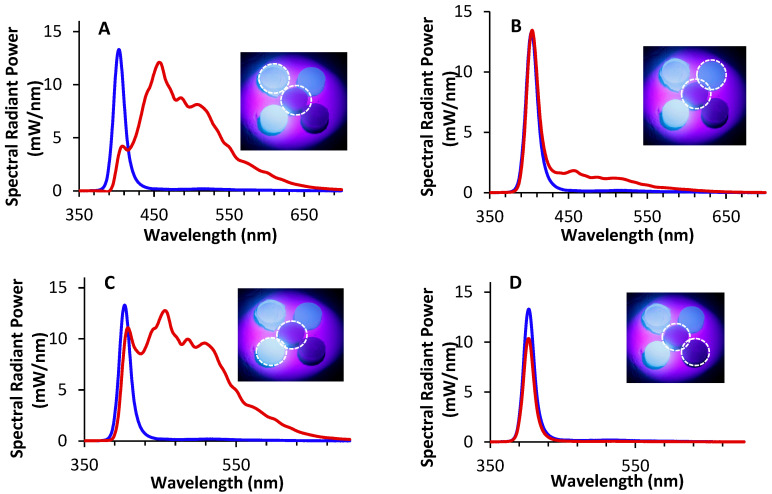
Four contrasting resin-based composites (RBCs) placed adjacent to one another with the enamel + dentin tooth sample in the center and exposed to UV light. The spectra for the RBCs is displayed by the red line adjacent to the spectra of the enamel + dentin tooth sample (blue line): (**A**) PowerFill monomer alone, (**B**) PowerFill monomer and filler, (**C**) Admira Fusion X-tra, and (**D**) Activa Bioactive Restorative.

**Figure 6 materials-17-04843-f006:**
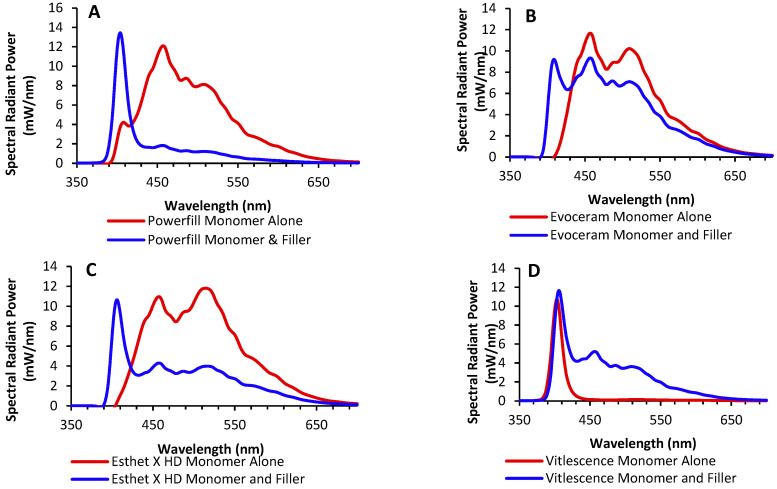
(**A**) The first Ivoclar sample (PowerFill) shows the absence in the unfilled and then the return of the opalescent peak in the filled product. (**B**) The second Ivoclar sample (Evoceram Monomer) shows the absence in the unfilled and the return of the opalescent peak in the filled product. (**C**) The Esthet X HD sample shows the absence in the unfilled and the return of the opalescent peak in the filled product. (**D**) Unfilled Vit-l-escence, did not follow the opalescent and fluorescent trends found in the unfilled versions of PowerFill, Evoceram, and Esthet X.

**Figure 7 materials-17-04843-f007:**
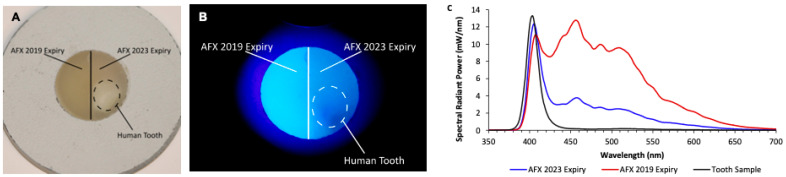
(**A**) Admira Fusion X-tra (AFX) samples with an expiry of 2019 and samples with an expiry date of 2023 under regular lighting plus the regular tooth sample. (**B**) AFX samples with an expiry of 2019 and expiry of 2023 under UV lighting, plus the regular tooth sample. (**C**) Spectrum for the 2023 expiry AFX, the 2019 expiry AFX, and the tooth.

**Table 1 materials-17-04843-t001:** The 46 RBCs categorized by their manufacturer.

RBC	Manufacturer	Shade
Filtek Bulk Fill Flowable (FBF)	3M, St. Paul, MN, USA	A2
Filtek One Bulk Fill Restorative (FO)	3M, St. Paul, MN, USA	A2
Filtek Universal (FU)	3M, St. Paul, MN, USA	A2
Filtek Z250 (Z250)	3M, St. Paul, MN, USA	A2
Aelite LS Posterior (ALP)	Bisco, Schaumburg, IL, USA	A2
Pyramid (P)	Bisco, Schaumburg, IL, USA	Dentin A3.5
Renamel Microfill (RM)	Cosmedent, Chicago, IL, USA	Enamel A1.5
Core Paste XP Syringeable (CP)	DenMat, Lompoc, CA, USA	White
Ceram X (CX)	Dentsply Sirona, Charlotte, NC, USA	A2
TPH Spectra HV (SHV)	Dentsply Sirona, Charlotte, NC, USA	A2
TPH Spectra ST HV (SSHV)	Dentsply Sirona, Charlotte, NC, USA	A2
TPH3 (TPH3)	Dentsply Sirona, Charlotte, NC, USA	A2
Esthet X HD (EXH)	Dentsply Sirona, Charlotte, NC, USA	A2
Allcem Veneer APS (AV)	FGM, Fort Lauderdale, FL, USA	Trans
Opus Bulk Fill (OB)	FGM, Fort Lauderdale, FL, USA	A2
G-aenial Universal Injectable (GUI)	GC, Lucern, Switzerland	A2
Kalore (K)	GC, Lucern, Switzerland	CVD
Venus Pearl (VP)	Heraeus, Hanau, Germany	A2
Durafill VS (DVS)	Heraeus, Hanau, Germany	A2
Charisma Syringe (CH)	Heraeus, Hanau, Germany	A2
Tetric EvoCeram (EC)	Ivoclar, Schaan, Liechtenstein	A2
Tetric EvoCeram Monomer (ECM)	Ivoclar, Schaan, Liechtenstein	IVW
Tetric EvoFlow Bulk Fill (EFB)	Ivoclar, Schaan, Liechtenstein	IVA
IPS Empress Direct (ED)	Ivoclar, Schaan, Liechtenstein	EA2
Tetric Prime (TP)	Ivoclar, Schaan, Liechtenstein	A3
PowerFill Monomer (PFM)	Ivoclar, Schaan, Liechtenstein	NA
Point 4 (P4)	Kerr, Brea, CA, USA	B1
SimpliShade (SS)	Kerr, Brea, CA, USA	Dark
Herculite Ultra (HU)	Kerr, Brea, CA, USA	Dentin A2
Clearfil Majesty Esthetic PLT (MEP)	Kuraray, New York, NY, USA	A2
Clearfil Majesty Posterior PLT (MPP)	Kuraray, New York, NY, USA	A2
CompCore AF SyringeMix Stack (CC)	Premier, Philedelphia, PA, USA	White
Activa Bioactive Restorative (AB)	Pulpdent, Watertown, NY, USA	A2
Aura Ultra Bulk Fill (AUB)	SDI, Bayswater, Australia	Universal
Estelite ∑ Quick (EQ)	Tokuyama Dental, Yamaguchi, Japan	A2
Omnichroma (OMN)	Tokuyama Dental, Yamaguchi, Japan	Universal
Mosaic (M)	Ultradent, South Jordan, UT, USA	A2
Vit-l-escence (V)	Ultradent, South Jordan, UT, USA	OW
Amelogen PLUS (AP)	Ultradent, South Jordan, UT, USA	A2
Transcend (TS)	Ultradent, South Jordan, UT, USA	Dentin A1
Grandioso (G)	Voco, Cuxhaven, Germany	A2
Arabesk Top (AT)	Voco, Cuxhaven, Germany	OA2
X-tra Fil (XF)	Voco, Cuxhaven, Germany	Universal
Admira Fusion Flow (AFF)	Voco, Cuxhaven, Germany	A2
Admira Fusion Xtra 2023 (AFX 2023)	Voco, Cuxhaven, Germany	Universal
Admira Fusion X-tra 2019 (AFX 2019)	Voco, Cuxhaven, Germany	Universal

## Data Availability

The raw data supporting the conclusions of this article will be made available by the authors on request.

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
