# Peer review of "Opalescence and Fluorescence of 46 Resin-Based Composites Exposed to Ultraviolet Light"

_materials, 2024, doi:10.3390/ma17194843_

Round 1

Reviewer 1 Report

Comments and Suggestions for Authors

This study examined both quantitatively and qualitatively the effect of using the resin-based composite (RBC) detection setting on the Woodpecker O Star curing light (Guilin Medical 68 Instrument Co.) on the opalescence and fluorescence from a wide range of RBCs. The hypothesis is that when exposed to UV light, all the RBCs will emit a similar level of opalescence and fluorescence. The paper should be revised.

-It should be more clearly described in abstract and in introduction why these investigations are important ?

-Chemical structures of resins of PowerFill (Ivoclar, 74 Schaan, Liechtenstein), EsthetX (DentsplySirona, Charlotte, NC, USA)) and Vit-l-escence 75 (Ultradent, South Jordan, UT, USA) should be provided.

-The title of figure 1 should be exact. The is a description for the title at the moment ?

-Practical value of the investigations is not clear and should be demonstrated?

-It should be described in conclusions, what new in the scientific field was found after the investigation ?

Author Response

Comment 1: It should be more clearly described in abstract and in introduction why these investigations are important ? 

Response 1: Thank you for pointing this out. We thought we had done so, but have further explained why this study is important.

It was concluded that because RBCs emit very different levels of opalescence and fluorescence, UV light from the Woodpecker O-Star cannot be relied upon to detect all brands of RBC on the tooth. The opalescence and fluorescence can also be used to detect changes in the formulation of the RBC.
 Comment 2: Chemical structures of resins of PowerFill (Ivoclar, Schaan, Liechtenstein), EsthetX (DentsplySirona, Charlotte, NC, USA)) and Vit-l-escence 75 (Ultradent, South Jordan, UT, USA) should be provided.
 Response 2: We do not agree with this because we did not examine the chemical structures of resins used in this study. It would require many articles to discern the chemical structures of these dental resins as their compositions are proprietary.    The paragraph now reads:

When isolating the contribution of various materials to fluorescence and opalescence, Figure 4 shows that it is impossible to make an overarching statement that UV light can detect all the RBCs on the market. Manufacturers use different resins and fillers in their RBCs and they do not fully disclose the proprietary compositions of the base resins or fillers they use in their RBCs. Thus, the use of the RBC detection mode from the Woodpecker O-Star light cannot be relied upon to detect all RBCs.  For example,  based on the evaluations of the three unfilled resins provided by Ivoclar, Dentsply and Ultradent, it appears that their proprietary resin monomers produce the fluorescence peaks under UV light in both the Ivoclar and DentsplySirona RBCs (Figure 6A, B, &C), but not for the Vit-l-escence resin from Ultradent, which produced less fluorescence than its filled version (Figure 6D). Adding inorganic filler in all the RBCs tested produced opalescence peaks under UV light. However, the fluorescence cannot be attributed to any one monomer (Bis-GMA, TEGDMA, etc.) or camphorquinone since the percentages of these compounds in the dental RBCs are unknown and are proprietary trade secrets. Future studies could use controlled amounts of various known monomers and fillers to identify the cause of fluorescence and opalescence in these experimental RBCs.

Comment 3: The title of figure 1 should be exact. The is a description for the title at the moment ?
Response 3: We have revised the title of Figure 1.

FIGURE 1. A) The distance of the resin-based composite sample was between 0-0.5 mm from the CC3 detector(behind). B) The Woodpecker O-Star curing light was fixed 20-mm above the RBC. The sample was centered under the UV spot C) Spectral radiant power (mW/nm) of the light from the Woodpecker O-Star. Emission peak at 402 nm. Beyond 435 nm, no light was detected.

 Comment 4: Practical value of the investigations is not clear and should be demonstrated?
Response 4: We have further emphasized the practical value of this paper.

One of the challenges when replacing these tooth-coloured RBC restorations is identifying the RBC on the tooth because RBCs are designed to match the color of the original tooth (8, 9). When only visual inspection is used by dentists to identify teeth with resin restoration, the accuracy and specificity are only 31.6% and 85%, respectively (11). To help the dentist identify the RBC on the tooth, some curing light manufacturers have included a composite detection mode in their curing light. This mode emits ultraviolet (UV) light (395 nm to 405 nm) to induce fluorescence of the RBC (12).  This has been reported to help the dentist detect the RBC on the tooth (12-14) and improves the detection accuracy to 92% and the specificity to 99% (11). But can the UV light from a dental LCU be relied upon to detect all RBCs?

Since no study has yet looked at a wide range of old and contemporary RBCs under controlled conditions, this study examined both quantitatively and qualitatively the effect of using the RBC  detection setting on the Woodpecker O Star curing light (Guilin Medical Instrument Co.) on the opalescence and fluorescence from a wide range of RBCs. The hypothesis is that when exposed to UV light, all the RBCs will emit a similar level of opalescence and fluorescence.

 Comment 5: It should be described in conclusions, what new in the scientific field was found after the investigation ?
Response 5:  Again, we thought we had already done so, but have repeated it here.

            The hypothesis for this study was rejected because the opalescence and fluorescence levels were markedly different between the RBCs when they were exposed to light from the Woodpecker O-Star curing light. Some RBCs could not be detected using this UV light.

The relative light emittance was lower for 13 RBCs compared to the dentin sample and 10 RBCs for the enamel plus dentin sample. However, a few RBCs were up to 6 times brighter than the samples of human teeth.

5 CONCLUSIONS

  1. When exposed to UV light from the Woodpecker O-Star curing light, the RBCs emitted very different levels of opalescence and fluorescence. Some RBCs could not be detected using this UV light.
  2. The filler used in the Ivoclar and DentsplySirona RBCs affects the opalescence peaks,

while the resin monomer influences the fluorescence peaks.

  1. Emission spectra from RBCs exposed to violet light can differ depending on when the RBC was manufactured. This can disclose changes that the manufacturer made to the composition of the RBC.

Reviewer 2 Report

Comments and Suggestions for Authors

The manuscript reports experimental measurements of opalescence and fluorescence of 46 resin-based dental composites (RBCs) exposed to UV light from a Woodpecker O-Star light control unit. By comparing with human dentin and enamel, it was found that some of the RBCs are less luminescent and some of the RBCs were much brighter. The results indicate that the commercially available RBCs emit very different levels of opalescence and fluorescence. In other words, caution is needed when using UV light to identify the boundary between RBCs and tooth structure. However, the study did not mention anything about dental adhesive layer, which is located in the boundary between RBCs and tooth in dental composite restorations. Consequently, the manuscript needs major revision before being accepted for publication in Materials.

Below are the specific comments from this reviewer:

1.      Figure 1 caption: B) The Woodpecker O-Star LCU was fixed…; Please provide the full term of LCU.

2.      Figure 4: Please remove the words of “13 RBCs less bright than the dentin specimen” for both A and B. Most of the RBSc shown in the figures are comparable or much more intense than the tooth specimen.

3.      The boundary materials for dental composite restorations are dental adhesives. The authors need justify why dental adhesive layer was not taken into consideration in this study.

Comments on the Quality of English Language

Minor editing of English language is required to improve the manuscript.

Author Response

Comment 1:   Figure 1 caption: B) The Woodpecker O-Star LCU was fixed…; Please provide the full term of LCU.
Response 1: We have revised Figure 1
FIGURE 1. A) The distance of the resin-based composite sample was between 0-0.5 mm from the CC3 detector(behind). B) The Woodpecker O-Star curing light was fixed 20-mm above the RBC. The sample was centered under the UV spot C) Spectral radiant power (mW/nm) of the light from the Woodpecker O-Star. Emission peak at 402 nm. Beyond 435 nm, no light was detected.

 Comment 2:  Figure 4: Please remove the words of “13 RBCs less bright than the dentin specimen” for both A and B. Most of the RBSc shown in the figures are comparable or much more intense than the tooth specimen. 

Response 2: We have revised the wording.

Figure 4: Ranking of the light emittance from the resin-based composites (RBCs) compared to the human tooth. A) Relative light emittance compared to a tooth sample composed mainly of dentin. B) Relative light emittance to a tooth sample composed mainly of enamel. The tooth sample is highlighted in red on each figure. The samples of the RBCs that were used in the blind ranking are highlighted in green.

  Comment 3: The boundary materials for dental composite restorations are dental adhesives. The authors need justify why dental adhesive layer was not taken into consideration in this study.
 Response 3: We do not agree with this point. Dentists and we are trying to discern between tooth and the composite, not tooth and bonding agent.  This would be an interesting study.

One of the challenges when replacing these tooth-coloured RBC restorations is identifying the RBC on the tooth because RBCs are designed to match the color of the original tooth (8, 9).

Comment 4: Minor editing of English language is required to improve the manuscript.
Response 3: We have checked the spelling and grammar to the best of our ability and used both Word and Grammarly as a final check.

Round 2

Reviewer 1 Report

Comments and Suggestions for Authors

Accept in present form

Reviewer 2 Report

Comments and Suggestions for Authors

In the revision, the authors have addressed most of the review comments.